# Definitions Matter: Guiding GPT for Multi-label Classification

**Youri Peskine**[*]
EURECOM
youri.peskine@eurecom.fr

**Damir Korenčić**[*]
Univ. Politècnica de València (UPV)
dkorenc@upvnet.upv.es

**Ivan Grubišić**
UPV
Ruđer Bošković Institute
grubisic@irb.hr

**Paolo Papotti**
EURECOM
paolo.papotti@eurecom.fr

**Raphael Troncy**
EURECOM
raphael.troncy@eurecom.fr

**Paolo Rosso**
UPV
prosso@dsic.upv.es

## Abstract

Large language models have recently risen in popularity due to their ability to perform many natural language tasks without requiring any fine-tuning. In this work, we focus on two novel ideas: (1) generating definitions from examples and using them for zero-shot classification, and (2) investigating how an LLM makes use of the definitions. We thoroughly analyze the performance of GPT-3 model for fine-grained multi-label conspiracy theory classification of tweets using zero-shot labeling. In doing so, we asses how to improve the labeling by providing minimal but meaningful context in the form of the definitions of the labels. We compare descriptive noun phrases, human-crafted definitions, introduce a new method to help the model generate definitions from examples, and propose a method to evaluate GPT-3's understanding of the definitions. We demonstrate that improving definitions of class labels has a direct consequence on the downstream classification results.

## 1 Introduction

Recent success of Large Language Models (LLMs) is due to their superior performance on various tasks (Qin et al., 2023; Liu et al., 2023; Yang et al., 2023), such as text generation (Dai et al., 2023; Khalil and Er, 2023), summarization (Wang et al., 2023), question answering (Peinl and Wirth, 2023; Tan et al., 2023), information retrieval (Omar et al., 2023), machine translation (Hendy et al., 2023; Jiao et al., 2023), and inductive reasoning (Wei et al., 2022; Kojima et al., 2022; Zhong et al., 2023). However, LLMs had much less success with solving specific tasks (Qin et al., 2023), such as text classification, where they still lag behind fine-tuned transformer models (Yang et al., 2023). Class definitions are commonly communicated to LLMs in the form of zero-shot prompts and few-shot exam-

ples, but both approaches are inferior to "defining" classes via a train set of annotated examples.

Misinformation has been a major research topic in the last few years, due to its significant impact on society. Social media have been a large vector of fake-news, especially during the COVID-19 pandemic. We revisit the problem of LLM zero-shot classification on the use-case of fine-grained multi-label conspiracy theory detection in tweets (Langguth et al., 2023), using GPT-3 model (Brown et al., 2020). We show that the challenging task of distinguishing conspiracy theories related to COVID-19 provides valuable insights into the capabilities and limitations of GPT-3. In particular, we aim at exploring the role played by the definition of the labels in zero-shot classification and explore the ability of GPT-3 in generating such definitions from a small set of labelled examples. Additionally, we propose a method to test the ability of GPT-3 to understand the definitions and apply them to classification[1].

## 2 Related Work

Zero-shot learning (Larochelle et al., 2008) is a machine learning problem where a model predicts classes without access to training data. It is based on the knowledge transfer from seen to unseen classes (Ye et al., 2020), using auxiliary information about the classes that can be provided to the model, such as text descriptions (Larochelle et al., 2008; Ba et al., 2015), semantic attributes (Xian et al., 2018; Lampert et al., 2013) or concepts defined in ontologies (Wang et al., 2018; Zhang et al., 2019a). Within the domain of natural language processing (NLP), especially text classification, the zero-shot learning approach has been used in a variety of tasks, such as argument (Zhang et al., 2021), keywords (Nam et al., 2016), intent (Liu

---

[*]Equal contribution

[1]The code is available at: https://github.com/dkorenci/gpt-def-zeroshot

et al., 2019) and utterance (Dauphin et al., 2013) classification.

Unlike the traditional approach for zero-shot text classification, LLMs are capable of solving zero-shot classification with no auxiliary information (Zhu et al., 2023; Törnberg, 2023; Kuzman et al., 2023). However, there is evidence that LLMs can outperform crowd workers in zero-shot text annotation tasks (Gilardi et al., 2023). A different approach is to fine-tune the LLM with examples made of manually crafted lists of dictionary/encyclopedia entries related to the target label (Gao et al., 2023) or to fine-tune smaller pre-trained language models with examples retrieved (Yu et al., 2023) or generated by LLMs (Zhang et al., 2023; Ye et al., 2022; Meng et al., 2022).

Approaches that use class definitions for constructing zero-shot classifiers and prompts have been proposed, both for CNNs and transformers (Zhang et al., 2019b; Yin et al., 2019), and for LLMs (Brown et al., 2020; Gilardi et al., 2023). However, they do not experiment with different variants of class definitions, but use one definition type for classification and report the results. Additionally, to the best of our knowledge, no previous research has used LLMs to generate a class-specific definition directly from examples and used that information for zero-shot text classification as we investigate in this work.

Semantic abilities of LLMs are commonly evaluated on NL understanding and reasoning tasks (Yang et al., 2023), but work on targeted evaluation of fine-grained semantic properties is scarce. Sahu et al. (2022) propose to evaluate the LLM's comprehension of query-related concepts by using a knowledge graph. To the best of our knowledge, there is no previous work focused on the ability LLMs to understand and apply definitions.

## 3   Methodology

In this section, we first describe the dataset used for conspiracy theory classification. Then, we present the approaches to definition-based zero-shot classification using GPT-3, and describe the method for evaluation of GPT-3's understanding of definitions. In this work, we use the term 'definition' to denote any additional explanation of the textual labels used to describe the class.

### 3.1   Conspiracy Theory Classification Dataset

The COCO dataset contains Twitter posts annotated w.r.t. 12 named COVID-19 related conspiracy theories (Langguth et al., 2023). Part of this dataset was used in MediaEval FakeNews challenge(Pogorelov et al., 2022), where participants had to detect mentions of 9 conspiracy theories[2] given the tweet texts. In this work, we use the MediaEval dataset, on which the best approaches used fine-tuned CT-BERT models (Müller et al., 2020; Peskine et al., 2021; Korenčić et al., 2022). The test set has not been shared publicly before early 2023, we can safely use it in our experiments as it is not part of the training data of GPT-3 that contains data up to September 2021[3].

### 3.2   Zero-shot Classification & Definitions

We leverage GPT-3 to perform multi-label zero-shot conspiracy theory detection on the test set of the MediaEval data. In particular, we perform binary classification for each conspiracy category, labeling each tweet as either mentioning the conspiracy or not.

Our baseline method relies on zero-shot (ZS) conspiracy theory classification from the textual label of the classes only (e.g. 'Anti-vaccination', 'Harmful Radiation', 'Satanism', etc). This assesses if the knowledge encoded in GPT-3 is able to differentiate between similar conspiracy theories.

Our next two approaches aim at improving the model's understanding of the label by providing more context in the prompt, specifically with a short definition of the label. We compare two types of definitions: Human-Written (HW) and Example-Generated (EG).

The HW definitions are given in the dataset overview paper (Langguth et al., 2023), and are part of the guidelines that were given to the human annotators of the data. Despite the definitions being well-written, annotators had to regularly discuss their understanding of the categories, suggesting the difficulty of the task at hand[4].

The EG definitions are generated with GPT-3 from the training set, by providing GPT-3 with 25 examples of tweets mentioning a given conspiracy

---

[2]Suppressed Cures (SUP), Behaviour Control (BHC), Anti-Vaccination (AVX), Fake Virus (FAK), Intentional Pandemic (INT), Harmful Radiation (HAR), Depopulation (DEP), New World Order (NWO), Satanism (SAT)

[3]According to https://platform.openai.com/docs/models/gpt-3-5

[4]The authors report a 92% inter-annotator agreement and more than half tweets had at least one disagreement.

theory and 25 examples of tweets not related to the conspiracy theory. We use 5 different random seeds to randomly select the example tweets[5], resulting in 45 definitions generated in total. We then ask the model to come up with a short textual description that could separate the sets of tweets. In this setting, we do not provide the textual label of the conspiracy theory, but we only give example tweets to the model. This prevents the model to rely on some of its pre-trained knowledge from reading the textual label. Examples of definitions which have been generated are in Appendix A.

For prompting the model we rely on simple prompts, using both OpenAI's 'system' and 'user' roles in our request. The 'system' message contains a description of the task, while the 'user' message contains the tweet's content to be classified. For the classification of the tweets, the definition is appended at the end of the 'system' message. Example prompts used to generate EG definitions and to annotate conspiracy theories are provided in Appendix B.

### 3.3 Definition Understanding

Approach of definition-based zero-shot classification leads to the question whether GPT-3 is able to correctly "interpret" definitions and "apply" them to text classification, which, in our case amounts to detection of conspiracy categories in texts. We propose two tests aimed at assessing if GPT-3 indeed "understands" the definitions given in the prompts.

The general approach is to use semantic similarity to measure how similarity between definitions correlates with the output of the definition-based classifiers, which we view as a result of GPT-3's "interpretation" and "application" of a definition. For example, one expectation is that similar definitions should lead to similar outputs. We perform the tests using the 45 example-generated definitions, which represent a challenging test case of mutually close definitions – randomly varied and derived from related categories. We define the semantic similarity of two definitions as cosine similarity of their embeddings, using state-of-art[6] sentence embedding model (Reimers and Gurevych, 2019).

The first test of GPT-3's "understanding" of the definitions measures whether EG definitions more

similar to HW ones guide the model to produce better classification results. This is achieved by correlating the similarity between the EG definitions and the corresponding HW ones, and the performance of the classifiers based on the generated definitions.

The second test measures whether mutually similar EG definitions guide the model to produce similar predictions. This is achieved by correlating the similarity between two EG definitions on one side, and the similarity of the corresponding classifiers' predictions on the other side. Similarity between two sets of predicted binary labels is calculated using Cohen's $\kappa$, a chance-corrected measure of annotator agreement.

## 4 Results

### 4.1 Conspiracy Theory Classification

In this section, we discuss the results of the different approaches on the classification of the full test set, totalling 823 tweets. Average results are in Table 1, and per-category results are in Figure 1. We use Matthews correlation coefficient, Precision, Recall and F1 score to compute the classification performance.

Results show that both EG and HW definitions outperform the ZS baseline. It supports the claim that GPT-3 is capable of leveraging the knowledge provided via the definitions to perform classification and, therefore, that definitions of the labels can be used to guide the model to better perform NLP tasks. While EG definitions do not reach the same performance as HW ones, they can still be used to significantly improve classification accuracy, especially in cases where the HW definition is not available. Our method shows that we can infer a textual description from examples and that GPT-3 can use it to better annotate future samples. Indeed, the usage of EG definitions leads to an average relative gain of around +10% in MCC, Precision and F1 scores compared to the ZS baseline. HW definitions see an even greater improvement of around +30% in average, showing the importance of a well-defined definition. However, these results are still far from the state-of-the-art CT-BERT fine-tuning methods.

Figure 1 reports the performances for all approaches per conspiracy theory. We observe a general trend with definitions having a positive impact on the performance for most conspiracy theories. However, some concepts are seemingly harder for

---

[5]Tweets in both sets can also support other conspiracy theories (multi-label classification problem)

[6]We use top-ranked `all-mpnet-base-v2` model: `https://www.sbert.net/docs/pretrained_models.html`

| Approach | MCC | Precision | Recall | F1 |
|---|---|---|---|---|
| Zero-shot | 0.398 | 0.331 | **0.852** | 0.440 |
| w/ Example-generated definitions | 0.442 | 0.371 | 0.831 | 0.485 |
| w/ Human-written definitions | **0.516** | **0.464** | 0.823 | **0.555** |
| CT-BERT ensembling | 0.780 | 0.779 | 0.849 | 0.810 |

Table 1: Performance of the LLM and transformer models using macro-averaging.

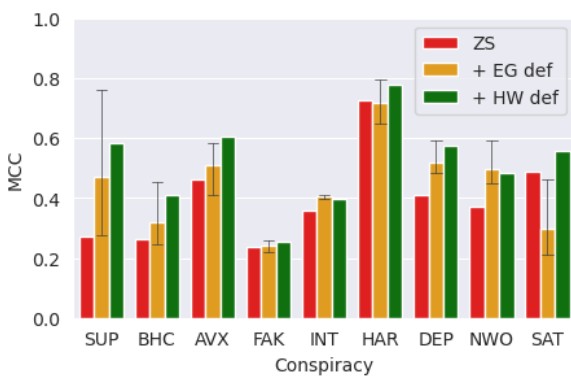

Figure 1: MCC score on the test set. Error bars show the minimum and maximum values (5 random seeds)

GPT-3 to produce useful definitions, such as Satanism, where the EG definitions lead to worse results than the ZS baseline. Also, some conspiracies are more robust to the EG definitions, as the variance is low and changing the samples lead to similar results, such as Intentional Pandemic, or Fake Virus. Lastly, some EG definitions lead to better results than the HW ones, suggesting that with a better sampling of the examples, this method could generate higher-performing definitions.

### 4.2 Definition Understanding Tests

The Spearman's rank correlation coefficients between semantic similarity of the definitions and the results of the definition-based zero-shot classifiers are shared in Table 2. The strength of the correlations is fair, which supports the claim that GPT-3 is able to correctly interpret the definitions and apply them to conspiracy detection. Namely, higher similarity between EG and HW definitions leads to more accurate classifications, which suggest that the model can translate better definitions into better predictions. Additionally, higher similarity between two EG definitions correlates with higher agreement between their corresponding predictions, which suggest that the model translates similar definitions into similar predictions.

An interesting question that stems from the variation of the definitions is whether the performance

increase is a result of the quality or the quantity of information in the definitions. To address this question we correlated the length of the 45 EG definitions measured by the number of tokens with their classification performance measured by MCC. We found a lack of correlation – a very small $\rho$ of 0.062. We take this as evidence supporting the claim that the performance depends on the quality, and not on the quantity, of information in a definition.

| | MCC | F1 |
|---|---|---|
| Similarity (EG, HW) | 0.375 | 0.390 |
| | Cohen's $\kappa$ | |
| Similarity (EG, EG) | 0.407 | |

Table 2: Results of the two definition understanding tests based on semantic similarity and classification results. Top row contains Spearman's correlations of similarity between EG and HW definitions, and performance of EG zero-shot classifiers. Bottom row contains correlations of similarity between pairs of EG definitions, and cohen's kappa of their classification.

## 5 Discussion & Future Work

While classification of conspiracy theories using EG definitions is done in a zero-shot fashion, the generation of the definitions still relies on annotated examples. This is different than standard in-context few-shot classification as these examples do not need to be part of the classification prompt. Indeed, it can be seen as a way to compress the information from few-shot examples into a shorter descriptive context that can be appended in the zero-shot prompt. Further experiments could explore this approach and compare it to standard in-context few-shot classification.

The correlation tests of definition understanding in Section 4.2 support the claim that GPT-3 can indeed interpret and apply the definitions correctly. This is complemented by the results in Section 4.1 which show that better definitions lead to better results. However, further testing should be done on

more LLMs and with other corpora. Such experiments are an interesting direction for future work with the potential to shed light on the semantic capabilities of LLMs.

The results of the definition-based zero-shot classifiers imply several practical recommendations and potential applications, all of which represent topics for future work. They include the use of the (high recall) classifiers to create more balanced samples for labeling, application of the classifiers to detect annotation errors, recommendations for mitigation of the low precision, and use of EG definitions for few-shot learning. More details on these topics can be found in Appendix C.

## 6 Conclusion

In this work, we analyze the impact of label definitions on the performance of GPT-3 zero-shot classification, on a challenging task of fine-grained conspiracy theory detection. We show that the use of better definitions leads to a significant gain in most metrics (MCC, Precision, F1). We also demonstrate an approach of generating definitions from examples. Human-Written definitions still provide better results, while example-generated definitions show promising performance. Additionally, we successfully tested GPT-3's ability to understand and apply these definitions for classification.

## Acknowledgements

This work has been partially supported by CHIST-ERA within the CIMPLE project (CHIST-ERA-19-XAI-003) and by ANR within the ECLADATTA project (ANR-22-CE23-0020). The work at the Universitat Politècnica de València is carried out in the framework of the XAI-DisInfodemics research project on eXplainable AI for disinformation and conspiracy detection during infodemics (Grant PLEC2021-007681), funded by MCIN/AEI/ 10.13039/501100011033 and by European Union NextGeneration EU/PRTR.

## Limitations

We conduct our experiments using only the GPT-3.5 model, which is not open sourced, and is accessible only as a cloud service which might incur high usage costs, and forces the users to rely on a third-party service.

The experiments are conducted on conspiracy theory classification, and thus the performance im-

provements might not be directly applicable to other multi-label classification tasks.

We mainly focus on providing better definitions for labels, but other factors, such as the quality of the input data, the prompt, and the model architecture can also play a role in improving classification performance.

The definition understanding analysis is based on several complex artifacts: LLMs, the model of semantic similarity, and the human-crafted definitions and annotations. Therefore, there might be other viable explanations of the obtained positive correlations, including unexpected interactions.

## Ethics Statement

Our study aims to improve the classification of conspiracy theories in social media posts, which might assist in detecting and mitigating misinformation, thus contributing to a more reliable and trustworthy online environment.

**Biases** We are aware of the biases of LLMs in classification tasks (Bender et al., 2021). Using language models to generate definitions might lead to unintended biases in the generated definitions, which could impact the performance of the classifier. However, our goal is to study the impact of definitions on the performance of GPT-3, which can bring insights on how to reduce bias with the right prompts and more diverse training data.

**Environmental Impact** The use of large-scale Transformers requires a lot of computations and GPUs/TPUs for training, which contributes to global warming (Strubell et al., 2020). This is a smaller issue in our case, as we do not train such models from scratch; rather, we fine-tune them on relatively small datasets or use models for inference in zero-shot settings.

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

# A  Examples of Definitions

## A.1  Definitions of Suppressed Cures Conspiracy Category

### A.1.1  Example Generated (Seed 0)

The definition of the concept is a conspiracy theory that suggests the existence of a deep state that is orchestrating the COVID-19 pandemic and blocking the release of Hydroxychloroquine, a cure for the virus. This theory also involves the belief that the pandemic is being used to push liberal agendas, create economic recession, help China's economy, and stop Trump rallies. It is often associated with the QAnon movement and involves the idea that Dr. Fauci is a Deep State Killer.

### A.1.2  Example Generated (Seed 1)

The definition of the concept is the use of hydroxychloroquine (HCQ) as a possible treatment for COVID-19.

### A.1.3  Human Written

Suppressed cures: Narratives which propose that effective medications for COVID-19 were available, but whose existence or effectiveness has been denied by authorities, either for financial gain by the vaccine producers or some other harmful intent.

## A.2  Definitions of Harmful Radiation Conspiracy Category

### A.2.1  Example Generated (Seed 4)

The definition of the concept is a conspiracy theory that claims that the COVID-19 pandemic is not caused by a virus, but rather by the implementation of 5G technology and that the pandemic is being used as a distraction to implement a digital currency and implantable microchips for population control. This theory has been widely discredited by scientific evidence and health organizations.

### A.2.2  Example Generated (Seed 1)

The definition of the concept is a conspiracy theory that claims that the COVID-19 pandemic is caused by the deployment of 5G wireless networks. This theory suggests that the radiation from 5G towers weakens the immune system, making people more susceptible to the virus, or that the virus is somehow being transmitted through the 5G network. Despite being debunked by scientists and health experts, this theory has gained traction among some individuals and has led to the destruction of 5G towers in some areas.

### A.2.3  Human Written

Harmful radiation: Narratives that connect COVID-19 to wireless transmissions, especially from 5 G equipment, claiming for example that 5 G is deadly and that COVID-19 is a coverup, or that 5 G allows mind control via microchips injected in the bloodstream.

## A.3  Definitions of Depopulation Conspiracy Category

### A.3.1  Example Generated (Seed 2)

The definition of the concept is depopulation, which refers to the deliberate reduction of the world's population through various means such as disease, vaccines, and other methods. The tweets in the first set suggest that there is a group of people, including Bill Gates, who are actively working towards depopulation as part of their agenda. The tweets in the second set, on the other hand, do not mention depopulation and focus on other topics related to COVID-19 such as conspiracy theories, mental health, and political issues.

### A.3.2  Example Generated (Seed 3)

The definition of the concept is depopulation, which refers to the deliberate reduction of the human population through various means such as dis-

ease, sterilization, or other forms of population control. The tweets in the first set suggest that some individuals believe that the COVID-19 pandemic and the development of vaccines are part of a larger depopulation agenda orchestrated by individuals such as Bill Gates. These beliefs are not present in the second set of tweets, which focus on other topics related to COVID-19 such as its impact on mental health, the effectiveness of vaccines, and political responses to the pandemic.

### A.3.3 Human Written

Depopulation: Conspiracy theories on population reduction or population growth control suggest that either COVID-19 or the vaccines are being used to reduce population size, either by killing people or by rendering them infertile. In some cases, this is directed against specific ethnic groups.

### A.4 Definitions of Satanism Conspiracy Category

#### A.4.1 Example Generated (Seed 0)

The definition of the concept is the belief in a conspiracy theory that the COVID-19 pandemic is part of a larger plan orchestrated by a group of powerful individuals or organizations, such as the Illuminati, the New World Order, or the Deep State, to exert more control over the population and ultimately lead to the implementation of the "mark of the beast," which could be a microchip or a vaccine that would be mandatory and used to monitor and control people's actions and movements. This belief often involves the idea that the pandemic is a hoax or exaggerated, and that the real agenda behind it is to push for a global government and depopulation.

#### A.4.2 Example Generated (Seed 3)

The definition of the concept is the belief that the COVID-19 vaccine or any other form of mandatory vaccination is the "Mark of the Beast" as described in the Book of Revelation. This belief is often associated with conspiracy theories involving the government, deep state, and Luciferian Freemasons who are seen as trying to control and enslave the population through the use of tracking chips and microchipped vaccines. The concept is rooted in religious and apocalyptic beliefs and is often used to justify opposition to vaccination and other public health measures.

### A.4.3 Human Written

Satanism: Narratives in which the perpetrators are alleged to be some kind of satanists, perform objectionable rituals, or make use of occult ideas or symbols. May involve harm or sexual abuse of children, such as the idea that global elites harvest adrenochrome from children.

## B Prompt Description

### B.1 Example Prompt for EG Definitions

SYSTEM = "You will be given two sets of tweets. The first set of tweets contains examples of texts that mention the same concept. The second set of tweets contains examples of texts that mention other concepts, but not the same concept that tweets from the first set. Your task is to provide the definition of the concept present in the first set"

USER = "First set of tweets:
[25x Tweets containing the conspiracy]

Second set of tweets:
[25x Tweets not containing the conspiracy]

Given those two sets of tweets, what is the definition of the concept present in the first set that is not present in the second set of tweets? Start your answer with: 'The definition of the concept is'"

### B.2 Example Prompt for annotating a Tweet with regard to a conspiracy theory

SYSTEM = "Your task is to label tweets regarding the '[CONSPIRACY]' COVID-19 conspiracy theory. The available labels are: 1) mentions the conspiracy, 2)

```
        does not mention the
        conspiracy .
The definition of the '[
        CONSPIRACY ] ' conspiracy theory
         is the following :
[CONSPIRACY definition ]"

USER = "[TWEET]

Does the tweet : 1) mention the '[
        CONSPIRACY ] ' conspiracy , 2) do
         not mention the '[CONSPIRACY
        ] ' conspiracy ? Please include
        the corresponding number in
        your answer ."
```

## C  Recommendations For Practical Use

In this section, we elaborate on some recommendations for applications of definition-based zero-shot classifiers. These recommendations are mainly motivated by the classification results from Section 4.1.

**Fixing the class imbalance for labeling**  Recall of the definition-based zero-shot classifiers is high and comparable to the recall of the fine-tuned model. Therefore, a possible application of such classifiers is the selection of text data for labeling, with the goal of fixing the class imbalance, i.e., increasing the expected proportion of positive examples. This approach could help mitigate the rarity of positive examples in many text classification use-cases, such as various misinformation detection scenarios.

**Correcting annotation errors**  Another potential application of the definition-based zero-shot classifiers is detecting and correcting annotation errors. The approach we propose is to perform error analysis of the classifiers based on human definitions, which are commonly used for text annotation. As suggested by low precision scores (see Table 1), the number of false positives is high – on average 145.89 texts per category for the test set of 830 texts. However, the number of false negatives is lower and more tractable (on average 27.11 texts per category). Additionally, high recall implies that the texts tend to be correctly detected as non-conspiracies, so the false negatives also seem more likely to identify examples wrongly annotated as conspiracies.

Our preliminary analysis indicates that this is indeed the case. We randomly selected 5 false negative texts per category and checked the annotations using the category definitions from Langguth et al. (2023). We found, on average, 3.8 labeling errors per category (76% of inspected texts).

**Mitigating the low precision**  The classification results in Table 1 show that the definition-based zero-shot classifiers suffer from low precision. This means that there is a high occurrence of false positives – texts belonging to other related categories being recognized as adhering to the definition of the category being classified. A possible remedy for this could be to upgrade the category definitions with text explicitly excluding similar categories.

**Example-generated definitions use cases for few-shot learning**  An interesting use-case of EG definitions is the fact that they serve as a way to encode a lot of information into a shorter paragraph. Indeed, the LLMs can provide a descriptive definition of the task from a set of examples. This way, rather than providing all the examples each time we want to annotate a sample, we can provide a much shorter context, allowing to reduce the prompt size, and thus the cost, significantly.

Also, the quality of the definition matters, meaning we can actually use a more powerful model (such as GPT-4) to generate the definition, but still use a cheaper model to run the annotation (such as GPT-3.5-turbo). This allows to annotate large amount of data with a higher-quality definition without increasing the cost by much.