# OpenReview forum: "Definitions Matter: Guiding GPT for Multi-label Classification"
_EMNLP/2023/Conference — EMNLP 2023 Findings_

### Official Review · Reviewer_hx9P · 2023-07-20

**Soundness:** 4

**Excitement:**

3: Ambivalent: It has merits (e.g., it reports state-of-the-art results, the idea is nice), but there are key weaknesses (e.g., it describes incremental work), and it can significantly benefit from another round of revision. However, I won't object to accepting it if my co-reviewers champion it.

**Paper Topic And Main Contributions:**

The authors describe a way to improve zero-shot multi-label classification of large language models in the domain of conspiracy theory. Especially, the authors suggest two low-laborious techniques that prompt more context information to overcome low performance in specialized classification tasks: i) Human-Written (HW) label definitions, and ii) example-generated definitions (EG). The latter is designed by providing the system with 25 examples each, containing a certain conspiracy theory or not. These approaches are compared to the zero-shot classification from class labels. Semantic correlation is documented between the HW definitions and EG, which in turn also are compared/correlated to the system output’s performance. All experiments are performed using GPT3.5


**Questions For The Authors:**

The correlations part interchangeably uses terms : Cohen’s kappa, Spearman’s correlation and Matthews correlation. Can you uniform that?

**Reasons To Accept:**

The paper is well-written, well-structured and clearly-formulated. That makes understanding easy also for people that do not work deeply in the LLMs domain. The offered approaches are complementing each other. They are derived reasonably from the lack of performance. However, the techniques also focus on low laborious work, hence, they are efficient.

It shows some good alternatives to time-consuming fine tuning and zero-shot problems.

**Reasons To Reject:**

I feel that the fact that the GPT version used is not open sourced and can incur usage costs a bit irritating.

**Reproducibility:**

3: Could reproduce the results with some difficulty. The settings of parameters are underspecified or subjectively determined; the training/evaluation data are not widely available.

**Reviewer Confidence:**

4: Quite sure. I tried to check the important points carefully. It's unlikely, though conceivable, that I missed something that should affect my ratings.

**Typos Grammar Style And Presentation Improvements:**

See questions, also Table 2 is not entirely clear. Why don't you provide an F1 for Cohen's kappa?

---

> ### Author Rebuttal · Authors · 2023-08-29
>
> We agree that closed models incurring usage costs is a limitation, albeit not a crucial one, as closed models should also be evaluated scientifically to the best of the ability of the academic community, in particular, when such models are proven to be effective in multiple benchmarks.
>
> GPT3.5 is a widely used state-of-art model that is commonly included as a benchmark, and many experiments are done solely with this model. To mitigate this issue and improve reproducibility, we provide all the outputs of GPT-3 including the generated definitions. The fact that the model is not open source is a limitation, but we point out that we are treating the model as a black box and that our evaluation methods do not require probing the model’s internals.
>
> We thank the reviewer for pointing out the potential unclarities about using Cohen’s kappa, Spearman’s correlation and Matthews correlation interchangeably. We note that Spearman’s Rho is used to measure correlation between (1) definition’s similarity and (2) either the zero-shot performance in EG\~HW (using Matthews correlation), or the similarity of classification outputs in EG\~EG (using Cohen’s kappa). These three metrics are distinct and used as described above.
>
> All the metrics in the table serve to compare two sets of classification labels. Top row contains standard classification performance metrics, since we compare classifiers’ labeling with the human gold labels. Bottom row compares labels produced by two classifiers and in this case without a reference label we think it is more correct to use a symmetric measure of inter-annotator agreement since there is no reference labeling.

---

### Official Review · Reviewer_rBbv · 2023-07-26

**Soundness:** 3

**Excitement:**

3: Ambivalent: It has merits (e.g., it reports state-of-the-art results, the idea is nice), but there are key weaknesses (e.g., it describes incremental work), and it can significantly benefit from another round of revision. However, I won't object to accepting it if my co-reviewers champion it.

**Paper Topic And Main Contributions:**

This paper explores the zero-shot capabilities of GPT-3 at the task of conspiracy theory detection in tweets about covid-19. (For instance, one conspiracy theory label used in the dataset used in the paper's evaluation is related to alternative cures being suppressed/hidden by the government.) The paper analyzes standard zero-shot performance --- where the model is only provided with the text input as well as a label (to predict yes/no) --- as well as scenarios where the model is also provided with descriptions/definitions of what the conspiracy category is about. For this latter setting, the paper explores handwritten definitions (provided by the creators of the dataset) as well as model-generated labels (generated after sampling several instances and asking the GPT model to describe definitions of each label category given these samples). The model finds that zero-shot performance is worse than fine-tuned model performance, and that the original handwritten definitions yield better zero-shot performance, which are not totally surprising results.

**Questions For The Authors:**

- Q1) When was the gpt-3.5-turbo-0301 model trained? This information would help our trust in the "Because the test set has not been shared publicly before early 2023, we can safely use it as it is not part of the training data of GPT-3. " (lines 126-128) statement.
- Q2) MCC, precision, recall, F1 are all metrics for binary classifiers. Do you use micro or macro averaging to aggregate the 12 mis-information types?
- Q3) Is the original COCO/MediaEval problem a 12-way classification task? Or do they do binary classification?
- Q4) I am confused at the "45 definitions generated in total" statement. Were the 5 random seeds each used to sample 25 tweets from each conspiracy label? Does one random seed yield 25 tweets, which in turn yields one definition?
- Q5) Do the tweets sometimes contain multiple conspiracy types? I wonder if this would effect the example-based generation of definitions.

**Reasons To Accept:**

- I appreciate the "Definition Understanding" analysis. This adds an extra layer to the paper beyond just model classification performance.
- Examples are provided in the Appendix, which were interesting to read.
- The scope of this paper is rather small, but I could see this type of paper being accepted to Findings or to a workshop.

**Reasons To Reject:**

- W1) The paper evaluates only on one dataset and on one task. Results and conclusions would be stronger if the analysis were applied to more datasets and more tasks.
- W2) Similarly, only one LLM model (GPT-3) is examined.
- W3) Some terms like "co-prediction" (line 278) and "in-context" (line 285) are not defined or explained.
- W4) A potential weakness is that the paper overlooks the presence of multi-label tweets. For instance, do multi-label tweets impact (i.e., harm) the effectiveness of the EG approach? I imagine that multi-label tweets would confuse the model insofar as the model would have a harder time generating a clear definition of a single label concept. See Q5 below.

**Reproducibility:**

4: Could mostly reproduce the results, but there may be some variation because of sample variance or minor variations in their interpretation of the protocol or method.

**Reviewer Confidence:**

3: Pretty sure, but there's a chance I missed something. Although I have a good feel for this area in general, I did not carefully check the paper's details, e.g., the math, experimental design, or novelty.

**Typos Grammar Style And Presentation Improvements:**

- "Recent success of Large Language Models (LLMs) is due to their superior performance" (line 021) sounds like circular reasoning.
- There are some various minor grammar mistakes. For instance I believe "Approach of definition-based..." should be "The approach of definition-based..." (on line 182). Similarly "as cosine similarity" should be "as the cosine similarity" on line 200.
- Inconsistent reference styles. For instance the references on lines 593-596, 601-604, 575-579, and 560-564 are all arXiv papers but have different reference formats. Similarly, only some references have URLs available.
- Section 5 mentions "in-context few-shot learning" but it is not clear what "in-context" refers to, nor is "few-shot" mentioned other than in the introduction.

---

> ### Author Rebuttal · Authors · 2023-08-29
>
> * W1 mentions that this work would benefit from using additional datasets and/or tasks. Indeed, this can be done in future follow-up works as the idea is not only tied to conspiracy theory. However, fine-grained multi-label conspiracy detection is a fitting task to experiment with generated definitions, as the definition of a conspiracy theory is not obvious and the difference between some conspiracies can be subtle (i.e. semantically similar). This paper is short one appropriate to describe findings per the ACL community tradition.
>
> * W2 states that the work could also benefit from using other LLMs. We experimented with GPT-3 as it was the best performing model to use out-of-the box. A similar trend was observed with GPT-4 on a smaller subset, but we did not scale the experiments because of the higher costs. Preliminary experiments with some smaller models (e.g. Bloom-7B, GPT-J) were not successful, as they were not able to differentiate between the 9 different conspiracies. We did not experiment further, but follow-up work could assess how other LLMs (e.g. Llama2, Falcon, Claude, etc.) handle definitions.
>
> * W3 highlights that the terms “co-prediction” and “in-context” are not defined in our work. By “in-context”, we mean the approach based on the LLM text interface that provides all the task- and example- related data as text prompt. Additionally, we define “co-prediction” as the agreement between two labeling results. In our case, we used Cohen’s kappa to measure the agreement between GPT-3 annotations given two different definitions. Our results suggest that the more similar the definitions are, the higher the agreement is, which means the more the co-predictions are aligned. The term “co-prediction” might be confusing, so we will edit the paper and refer to Cohen’s Kappa instead.
>
> * Q1 asks when was the gpt-3.5-turbo-0301 model trained. According to OpenAI API documentation (https://platform.openai.com/docs/models/gpt-3-5), this model was trained on data up to September 2021. Since the test set of MediaEval has only been released publicly in early 2023, the model should not have seen the data during training.
>
> * Q2 refers to how we averaged the metrics to aggregate the misinfo-types. We use macro-averaging to aggregate the 9 classes. This correction will be edited in the paper.
>
> * Q3 requests more information about the original datasets. Both COCO and MediaEval are multi-label multi-class datasets. They annotate each tweet with a 3-way label for 12 and 9 classes respectively. The original datasets are not binary and have two different labels for “discussing” a conspiracy and “promoting” it; we merged both labels to just consider tweets “mentioning” the conspiracy. COCO is an upgraded version of MediaEval dataset that contains 9 original conspiracy categories and augments the taxonomy with additional categories: a rare label "Esoteric misinformation", "other conspiracies" and "other misinformation". In our work, we opted for the MediaEval dataset as it contains all the relevant conspiracy-related labels. These clarifications will be edited for the final version of the paper.
>
> * Q4 points out confusion about the number of generated definitions and the seeds to pick examples. To clarify, one seed is used to pick the 50 random examples, 25 of which refer to the conspiracy and 25 of which do not. These examples are used to generate a definition. Because there are 9 classes in the MediaEval dataset and we generate definitions with 5 random seeds, this equals to 45 generated definitions.
>
> * W4 and Q5 refer to the presence of multi-label tweets in the data and how it can impact the generation of the definitions. Indeed, most tweets contain multiple conspiracy types. This definitely has an impact on the quality of the generated definition. However, it is not clear what strategy would yield the best results: to only include tweets that mention a single conspiracy theory, or to select tweets with multiple conspiracies and sample tweets mentioning the same other conspiracies in the negative examples. We decided to randomly select the examples on 5 different seeds to assess the variance of the generated definition.
>
> We thank the reviewer for pointing out the grammar mistakes and the inconsistencies in reference styles. These will be fixed in the final version.

---

### Official Review · Reviewer_u3c7 · 2023-08-02

**Soundness:** 3

**Excitement:**

2: Mediocre: This paper makes marginal contributions (vs non-contemporaneous work), so I would rather not see it in the conference.

**Paper Topic And Main Contributions:**

The article asses how to improve the labeling by providing minimal but meaningful context in the form of the definitions of the labels.

**Questions For The Authors:**


I haven't conducted a thorough literature review, but I find it a bit surprising the claim that this is the first study to examine the handling of definitions, especially when in the following section, it defines "definition" as the description of classes. Are there truly no existing works on zero-shot learning where the language model is introduced to class definitions?

The analysis of similarity between EG and HW definitions and their correlation with effectiveness is interesting. Even though this is more of an engineering-focused article than a scientific one, at least the analysis regarding the impact of definitions provides very reusable knowledge. However, I'm not sure if the second experiment holds much interest, as it seems obvious that the more the model's input resembles the output. The overall comparison between approaches with and without definitions is also intriguing. Nevertheless, it's not clear to me what the alternative to definitions would be. That is, it's understood that generally, providing more information in the prompt leads to more effective responses. In this article, the authors focus on definitions. But is there any information that can be added to the prompt unrelated to definitions? To what extent are we simply confirming the phenomenon of the benefit of adding more problem-related information to the prompt in general?

I have no objections regarding the formal aspects of the article. I believe it is well-written, and the examples are illustrative. Furthermore, the state of the art is quite comprehensive. My only concern pertains to the significance of the contributions.



**Reasons To Accept:**

- The methodology is appropriate.
- The literature review is comprehensive.
- The conclusions about the effect of adding label definitions is useful.
- The paper is easy to follow.

**Reasons To Reject:**

- Some doubts about the paper contribution. No alternative overspecification of the prompt is studied apart from the label definitions.

- The second experiment (similarity between EG vs. output) is obvious.



**Reproducibility:**

4: Could mostly reproduce the results, but there may be some variation because of sample variance or minor variations in their interpretation of the protocol or method.

**Reviewer Confidence:**

4: Quite sure. I tried to check the important points carefully. It's unlikely, though conceivable, that I missed something that should affect my ratings.

---

> ### Author Rebuttal · Authors · 2023-08-29
>
> In this work, we focus on examining how GPT-3 uses label definitions for zero-shot classification. We also note other possibilities of study, such as few-shot prompting, in the paper. To keep the paper short and focused, we do not perform an exhaustive examination of prompting strategies, but we rather perform an investigation of the handling of definition-based prompts by a LLM.
>
> We do not claim that “this is the first study to examine the handling of definitions”. Within the general context of “definition handling”, approaches that use class definitions for constructing zero-shot classifiers and prompts have been proposed. Such approaches can be based on traditional language models and transformers [1,2] or LLMs [3,4]. However, they do not experiment with variants of class definitions, but use one approach for classification and report the results. We thank the reviewer for this comment, since this line of related work will add context and clarity to the paper, and emphasize the novelty of our approach.
>
> [1] *Integrating Semantic Knowledge to Tackle Zero-shot Text Classification* (Zhang et al., NAACL 2019)
>
> [2] *Benchmarking Zero-shot Text Classification: Datasets, Evaluation and Entailment Approach* (Yin et al., EMNLP-IJCNLP 2019)
>
> [3] *Language Models are Few-Shot Learners* (Brown et al., NEURIPS 2020)
>
> [4] *ChatGPT Outperforms Crowd-workers for Text Annotation Tasks* (Gilardi et al., Proceedings of the National Academy of Sciences, 2023)
>
>
> As we stated in the paper, we focus specifically on (1) generating definitions from examples and using them for zero-shot classification, and on (2) investigating how a LLM makes use of the definitions. The latter goes beyond definition-based classification and uses the classification results as a proxy for gauging definition handling. We have claimed that (1) and (2) are novel ideas. We agree that this should be formulated more clearly and we thank the reviewer for pointing out this confusion.
>
> Results may not be surprising - more similar definitions lead to more similar classifications, statistically. However, considering the complexity of GPT-3 and current level of its understanding, we argue that systematically testing its abilities to correctly apply definitions is useful. Specifically, we interpret the correlation as an indicator of the ability of GPT3.5 to interpret definitions correctly. Moreover, we construct an experiment that can be used to measure the correctness of the definition application, confirm a model property in a systematic manner, and provide a benchmark for other LLMs.
>
> It should be noted that the two correlation experiments (EG ~ HW, EG ~ EG) are similar and serve the same purpose: to gauge the quality of the definition handling. The difference is that in the EG~HW experiment, one of the definitions (HW) and the associated (gold) labels are fixed. But the idea of the test is the same - a semantically similar definition should, if applied correctly, lead to similar classification results.
>
> While our experiments confirm that adding more problem-related information to the prompt is beneficial for the classification, we are primarily concerned by the quality of information, as opposed to the quantity of information. Analogously to what the reviewer correctly pointed out about the quantity of information, we also believe that it is generally understood that adding higher quality information leads to better prompts. The experiments in our view confirm this expectation quantitatively in a systematic manner. We mention the alternative ways to add information: examples, and problem descriptions, but focus solely on the label definitions in the context of zero-shot classification.
>
> Inspired by the previous discussion we correlated the definitions’ lengths (num. tokens) with the classification performance in the EG~HW experiment, using the length as the approximation of the quantity of information in the definition. We found a lack of correlation: very small Rho of 0.062 and a large p-value of 68%. More specifically, the probability of this level of correlation is 68% under the null hypothesis of there being no correlation. This can be interpreted as evidence that the performance quality, ie, content, of the definition matters, and not its length. This interesting datapoint will be added to the paper, but we provide it here primarily for the purpose of discussion, and for clarity.

---

### Meta-Review · Area_Chair_Uz4f · 2023-09-20

**Recommendation:** 3

**Metareview:**

This paper presents a simple prompting strategy for zero-shot classifying tweets according to COVID-19 conspiracy theories. Examples are augmented with a human-written or generated definition of the conspiracy theory being classified. The added definitions (especially human-written and generated ones similar to human-written ones) improve accuracy at the task. Reviewers generally thought the method is well-motivated and the paper is clear and well structured. One concern is the narrow set of experiments (one task and only using ChatGPT), but I think this lack of breadth is acceptable, although not ideal, for a short paper. The authors address reproducibility concerns over ChatGPT in the rebuttal. Reviewers did not give high excitement scores because the method is simple and the results are mostly unsurprising. However, that does not prevent the method from being useful or worth studying, and the paper does provide a focused contribution suitable for a short paper.

---

### Decision · Program_Chairs · 2023-10-07

**Decision:**

Accept-Findings

**Comment:**

This paper presents a simple prompting strategy for zero-shot classifying tweets according to COVID-19 conspiracy theories. Examples are augmented with a human-written or generated definition of the conspiracy theory being classified. The added definitions (especially human-written and generated ones similar to human-written ones) improve accuracy at the task. Reviewers generally thought the method is well-motivated and the paper is clear and well structured. One concern is the narrow set of experiments (one task and only using ChatGPT), but I think this lack of breadth is acceptable, although not ideal, for a short paper. The authors address reproducibility concerns over ChatGPT in the rebuttal. Reviewers did not give high excitement scores because the method is simple and the results are mostly unsurprising. However, that does not prevent the method from being useful or worth studying, and the paper does provide a focused contribution suitable for a short paper.